# The relationship between family environments growing up and behavioral health among LGBTQ+ adults: The mediating role of internalized homonegativity

**Pin-Chen Chiang**[1*], **Yinuo Xu**[1,2], **Denise Yookong Williams**[3], **Ankur Srivastava**[1], **Jake A. Leite**[1], **Adam R. Englert**[1], **William J. Hall**[1]

**1** School of Social Work, University of North Carolina at Chapel Hill, Chapel Hill, North Carolina, United States of America, **2** School of Education, University of North Carolina at Chapel Hill, Chapel Hill, North Carolina, United States of America, **3** School of Social Work, College of Health Sciences, University of Missouri-Columbia, Columbia, Missouri, United States of America

* pcchiang@unc.edu

## Abstract

### Purpose

Disparities in the behavioral health outcomes for lesbian, gay, bisexual, transgender, and queer+ (LGBTQ+) adults—such as depression, anxiety, and alcohol use—are often attributed to experiences of discrimination, victimization, and lack of supportive environments, including hetero- and cis-normative family settings. Yet, how family environments in childhood influence LGBTQ+ adults' behavioral health and internalized homonegativity has not been extensively examined.

### Methods

This study utilized a U.S. national dataset of LGBTQ+ adults ($N = 499$). Data were collected in November 2020 using an online survey. A series of multivariate ordinary least squares regression models and Sobel tests were performed.

### Results

Results showed that as homophobic messages from family increased, levels of depression ($\beta = .19$, $p < .001$), anxiety ($\beta = .17$, $p < .001$), and internalized homonegativity ($\beta = .13$, $p < .01$) increased. As the conservatism level in families increased, levels of alcohol use ($\beta = .11$, $p < .05$), and internalized homonegativity ($\beta = .15$, $p < .01$) increased. In the mediation analysis, internalized homonegativity was found to partially mediate the relationship between homophobic messages from family and depression ($z = 3.35$, $p < .001$), homophobic messages and anxiety ($z = 3.09$, $p < .01$), and conservatism in family and alcohol use ($z = 2.80$, $p < 0.01$). Internalized homonegativity also fully mediated the relationship between homophobic messages

**Data availability statement:** All relevant data are within the manuscript and its Supporting Information files.

**Funding:** Research reported in this article was supported by the National Institute of Minority Health and Health Disparities of the National Institutes of Health (https://www.nimhd.nih.gov/) under award number R21MD012687. W.H. is the author who received this award. The funders had no role in study design, data collection and analysis, decision to publish, or preparation of the manuscript.

**Competing interests:** The authors have declared that no competing interests exist.

and alcohol use ($z = 2.66$, $p < 0.01$), conservatism in family and depression ($z = 3.76$, $p < 0.001$), and conservatism in family and anxiety ($z = 3.45$, $p < 0.001$).

## Conclusion

Study findings underscore the importance of inclusive climates within a family and internalized homonegativity as a mediator for LGBTQ+ individuals' behavioral health. Implications for intervention and future research are discussed.

## Introduction

Lesbian, gay, bisexual, transgender, queer and other sexual and gender diverse (LGBTQ+) adults experience heightened rates of negative behavioral health outcomes (e.g., depression, anxiety, alcohol use) compared to their heterosexual and cisgender peers [1–3]. These disparities are often attributed to the presence of unique and chronic stressors, referred to as "*minority stress*"—that may include experiences of discrimination, victimization, and lack of supportive environments with pervasive heteronormativity and cisnormativity [4,5]. Heteronormativity is the assumption that heterosexuality is the only "normal" sexual orientation, often leading to the privilege of heterosexual relationships and gender norms within society. Likewise, cisnormativity views cisgender as the only acceptable gender identity and confers privileges and advantages to individuals whose identities fall within its imposed constraints. Even though societies adopt more policies protecting LGBTQ+ rights and discussing LGBTQ+ issues openly more than before, heteronormativity and cisnormativity remain dominant in the countries worldwide and affect social systems and LGBTQ+ people's health and development [6,7]. For example, social norms and expectations on gender and sexuality create additional challenges for LGBTQ+ populations in healthcare settings in terms of access to information or quality care [8,9]. In addition, laws that criminalize homosexuality or restrict the rights of transgender and gender-diverse individuals can place LGBTQ+ people at increased risk of physical violence, job loss, social exclusion, and suicidality [10,11]. Among social systems, the family environment has a particularly direct and profound influence in shaping and reinforcing cis- and heteronormativity on LGBTQ+ individuals [12]. In the United States, only 38% of LGBTQ+ young people reported home as a LGBTQ+-affirming space [13]. Childhood family environments, which may involve experiences of power imbalances and lack of autonomy, can have lasting effects on behavioral health, extending into adulthood [12,14–16].

An issue closely related to behavioral health is internalized homonegativity (IH), a process in which LGBTQ+ individuals receive negative social messages about their sexual identity and incorporate those views into their self-concept [17,18]. IH is also considered a consequence of heteronormative societal beliefs and affects one's behavioral health [4,19,20]. Yet, how the family environment in childhood influences LGBTQ+ adults' behavioral health and whether IH plays a mediation role have not been extensively examined. This study aims to add to this literature using a national survey of LGBTQ+ adults.

## Behavioral health disparities facing LGBTQ+ adults

Compared to heterosexual and cisgender people, LGBTQ+ populations are disproportionately struggling with mental health problems and suicidality [1,21]. From U.S. Census Bureau data, 18.8%–60.8% of LGBTQ+ adults (depending on subgroup) reported symptoms of anxiety, compared to 14.3%–42.5% of non-LGBTQ+ peers [2]. Additional data showed that compared to their non-LGBTQ+ counterparts, LGBTQ+ adults report higher symptoms of depression (11.4%–28.9% vs 15.5%–50.0%, respectively) and major depressive episodes (7.0% vs 26.7%, respectively [2,3]. Additionally, LGBTQ+ adults were more likely to engage in binge drinking alcohol during the past 30 days compared to their non-LGBTQ+ counterparts (32.0% vs 22.0%) [3]. Given the behavioral health disparities facing LGBTQ+ populations, it is essential to understand the mechanisms behind and factors that could potentially mitigate the disparities.

## Social and family environments for LGBTQ+ people

Affirming and inclusive social environments for LGBTQ+ people have multiple characteristics, including acknowledging LGBTQ+ identities, providing safe spaces, using inclusive language, and confronting stigma and discrimination [22,23]. An LGBTQ+-affirming or inclusive environment is a safe space for individuals expressing themselves and seek help or support [24,25]. Researchers have investigated the influences of state, school, and community climates among LGBTQ+ populations and found that more affirming environments are associated with lower odds of binge drinking, depression, and suicidal ideation among LGBTQ+ youth [24,26–31]. Alternatively, spaces where behaviors such as making negative comments about LGBTQ+ issues, using derogatory slurs for LGBTQ+ people, enforcing gender-binary norms, not welcoming LGBTQ+ people to family activities, or espousing hostile political or religious views characterize non-affirming or rejecting environments for LGBTQ+ people [32–34]. These non-affirming or hostile environments could fall under the category of sexual orientation microaggressions and can negatively affect LGBTQ+ people's behavioral health [35,36].

Research on family environments has explored relationships between parenting practices, acceptance or rejection responses to children's coming out, and parental cultural expectations of children [37–41]. Despite important findings about influences of family and caregiver interactions with LGBTQ+ children's behavioral health outcomes, existing studies largely focus on individual-level factors. For example, parental responses to coming out, family member's acceptance-rejection responses, family connectedness, and parents' awareness of children's identities are often discussed in literature, and they predict LGBTQ+ people's behavioral health outcomes [15,42]. Broader influences of family views, interactions, and beliefs in the childhood environment and their relationships with LGBTQ+ behavioral health outcomes in adulthood have been under-researched but may play important roles in understanding and improving LGBTQ+ behavioral health disparities. Next, we discuss various domains of the family environment for LGBTQ+ people and their influences on LGBTQ+ behavioral health.

**Homophobic language used in family.** Not only do overt acts of homophobia cause harm, but also subtle forms of homophobia, such as hearing homophobic language in social environments, can threaten the psychological safety and well-being of LGBTQ+ youth [12,43,44]. Prejudice-based words can impact stigmatized groups even if they are not the direct target of the comments [45]. Silverschanz and others found that college students in the northwestern U.S. who experienced ambient heterosexist harassment, such as hearing offensive remarks about LGBTQ+ people, reported higher substance use [46]. Those exposed to both personal and ambient harassment had worse well-being than those unexposed. The study highlighted the harmful impact of homophobic language. Other studies also examined the associations between exposure to homophobic language and various outcomes including psychological distress, and found a negative impact from indirect discrimination [44,45]. Current literature predominantly represents rejecting responses from families to LGBTQ+ children through explicit verbal harassment [33,34,47]. However, environmental factors also play a crucial role in shaping these responses. The mechanisms of ambient homophobic messages within a family and its subsequent impacts on LGBTQ+ youth behavioral health are not yet fully understood.

**Political conservatism in family.** Political discourse is another way heteronormativity can manifest in social settings. Historically, LGBTQ+ rights have been supported by liberal political groups and opposed by conservative political groups [48,49]. Studies suggest that people with conservative political ideology generally have more negative attitudes toward LGBTQ+ policies, less willingness to engage in LGBTQ+ activities, and greater endorsement of binary gender beliefs [50–54]. Research suggests that conservative family members oftentimes view youth sexual and/or gender expressions and identities that deviate from cis-heteronormative norms as temporary and expect individuals to settle into heterosexual relationships over time [55–57]. In fact, some caregivers self-identify as progressive but still grapple with accepting LGBTQ+ identities within their family unit [58]. Familial perspectives and pressures aligned with heteronormative societal views and practices generally inhibit LGBTQ+ youth development and may exacerbate minority stressors, such as identity management and concealment, subsequently adversely impacting mental well-being [59,60]. Negative LGBTQ+ political attitudes can contribute to psychosocial harm to LGBTQ+ individuals, as evidenced by numerous studies elucidating the deleterious impacts of recent increasing anti-LGBTQ+ policies on LGBTQ+ youth families and mental well-being [10,61–63].

**Openly LGBTQ+ people in family.** The composition and outness of family members also contribute to the family context for LGBTQ+ youth. Research suggests that growing up with LGBTQ+ parents can offer benefits for LGBTQ+ children, such as a more positive coming-out process and reduced concerns about family rejection [64]. In addition, extended relatives who identify as LGBTQ+ , such as aunts or cousins, were regarded as important supports for LGBTQ+ young adults navigating family environments centering heteronormativity [65]. During adolescence, LGBTQ+ youth often look to family members as role models [66], and adolescents seek psychological support, shared experience, self-acceptance, and assistance in overcoming challenging experiences from role models [67]. Therefore, open LGBTQ+ adults in the family could serve as both role models and sources of social supports to help foster positive development and well-being of LGBTQ+ individuals However, the impacts of having openly LGBTQ+ people within a family unit while growing up and the subsequent influences on LGBTQ+ individuals' behavioral health in adulthood has not been adequately investigated.

### Internalized homonegativity as a mediator

Underlying psychological processes can be important mechanisms in behavioral health disparities with marginalized populations. In Hatzenbuehler's *Psychological Mediation Framework* (PMF) [68], cognitive processes are one of the mechanisms mediating the relationship between stigma-related stress and psychopathology. That is, negative views of the self may result from stigma-based stressful environments and contribute to behavioral health outcomes. IH is a psychological process of absorbing negative social views into the self-concept, often resulting in the development of an internal conflict with one's own identity [18]. IH stems from stigma-based stress exposure in hostile environments [69,70] and is a predictor of poor behavioral health outcomes [18,71–73]. Studies have shown that IH plays a mediation role between cultural expectations and past parental rejection and current psychological distress among LGBTQ+ population [38,74]. However, again, the pathways from the family environmental factors through IH to behavioral health were not fully examined yet.

### Current study

With few studies focusing on the influences of family environment factors on LGBTQ+ adult's behavioral health, the purpose of this study was to understand whether family environment factors growing up are risk or protective factors for LGBTQ+ adult behavioral health problems, including depression, anxiety, and alcohol use. In addition, the potential mediating role of IH in the path between family factors and outcomes will be examined. This study aims to answer the following research questions: (1) How are family environment factors growing up (i.e., homophobic messages received from family; conservatism level in family growing up; having openly LGBTQ+ people in family or not) associated with LGBTQ+ adult children's behavioral health issues and IH? (2) How does IH mediate the relationships between family environment and

the behavioral health outcomes of depression, anxiety, and alcohol use? Our analyses were guided by this conceptual model (Fig 1), which is based on prior theory and research as described above.

## Methods

This study used cross-sectional data from a national U.S. study of LGBTQ+ adults. Data were collected in an online survey from November 2, 2020 to February 2, 2021 through CloudResearch, a participant-sourcing platform for online research [75]. Informed consent was attained online by the adult participants anonymously agreeing to participate, and then participants completed survey questions about demographics, IH, behavioral health, including depression, anxiety, and alcohol use, and family environments growing up. Participants were paid $10 for their participation. Validity check items were embedded in the survey, and mischievous respondents were removed. The study was approved by the authors' Institutional Review Board (Reference: R21MD012687).

### Measures

**Internalized homonegativity.** The Internalized Homonegativity Inventory (IHNI) is a 23-item scale with three subscales that include IH, morality of homosexuality, and identity affirmation [76]. IHNI responses were measured with a 7-point Likert-type scale from 1 (Strongly Disagree) to 7 (Strongly Agree) to understand the emotions and thoughts related to being a sexual minority. Example IHNI items included the following: "I feel ashamed of my homosexuality." "In my opinion, homosexuality is harmful to the order of society." "I am thankful for my sexual orientation." The subscale of identity affirmation was reverse scored. Average scores ranging from 1 to 7 were used for analysis. Higher scores indicate higher levels of IH. In this study, the internal consistency reliability of IHNI was very good ($\alpha = .94$).

**Depression symptoms.** The Clinically Useful Depression Outcome Scale (CUDOS) is a 16-item scale of depressive symptoms [77]. Participants' indicated the frequency that they had experienced depressive symptoms in the past 2 weeks using response options that ranged from 1 (Never) to 5 (Almost Every Day). Average scores ranging from 1 to 5 were used for analysis. Higher scores indicate higher levels of depressive symptom severity. In this study, the internal consistency reliability of the CUDOS was very good ($\alpha = .93$).

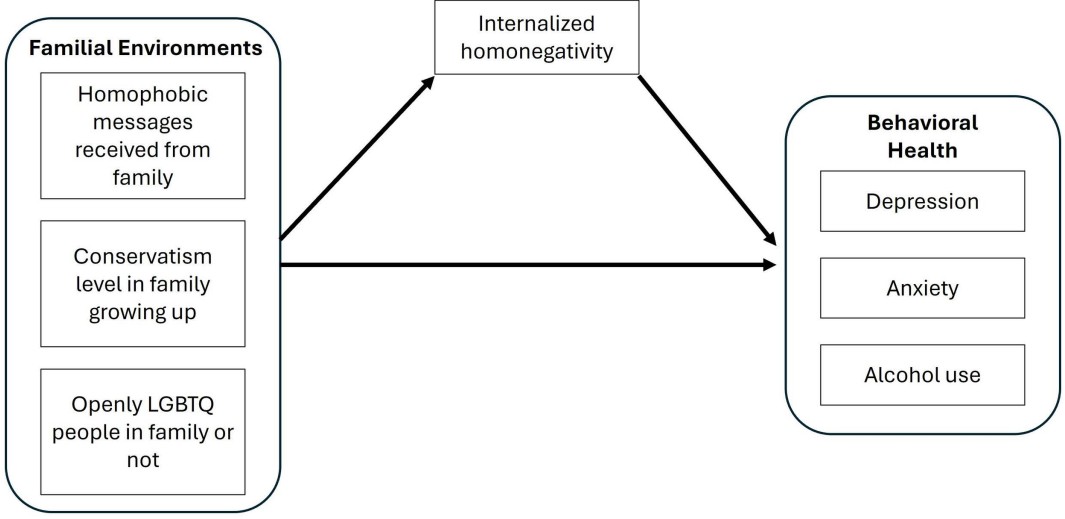

**Fig 1. Conceptual model.**

**Anxiety symptoms.** Anxiety symptoms in this study were measured with Beck Anxiety Inventory (BAI) [78]. The BAI consists of 21 anxiety symptom items. Participants rated how much they had been bothered by each symptom in the past 30 days using response options that ranged from 1 (Not At All) to 4 (Extremely). Average scores ranging from 1 to 4 were used for analysis. Higher scores indicate higher levels of general anxiety symptom severity. In this study, the internal consistency reliability of the BAI was very good (α = .95).

**Alcohol use.** PROMIS Item Bank v1.0 - Alcohol: Alcohol Use – Short Form 7a was a 7-item, 5-point Likert-type scale [79]. Participants' responses ranging from 1 (Never) to 5 (Almost Always) to answer alcohol use statements in the past 30 days. Example items include "I spent too much time drinking alcohol" or "It was difficult for me to stop drinking after one or two drinks." Average scores ranging from 1 to 5 were used for analysis. Higher scores indicate higher levels of alcohol use. In this study, the internal consistency reliability of the Alcohol Use – Short Form 7a was very good (α = .97).

**Homophobic messages received from family.** Homophobic messages received from family is a 5-item measure with a 5-point Likert scale response set, which was modified from the Attitudes Toward Lesbians and Gay Men Scale (ATLG-R-S5) [80]. Participants chose 1 (Never) to 5 (Almost Always) to answer the frequency they heard homophobic messages from their family growing up. Example items include: "Sexual activity between people of the same sex is just plain wrong" or "LGBTQ+ people are disgusting." Average scores ranging from 1 to 5 were used for analysis. Higher scores indicate the higher frequency of receiving homophobic messages from family growing up. In this study, the internal consistency reliability of ATLG-R-S5 was good (α = .87).

**Conservatism level in family growing up.** Conservatism level in one's family growing up was measured with a 1-item, 9-point Likert scale question with options from 1 (Very Liberal) to 9 (Very Conservative). Higher scores indicate higher level of conservatism in family.

**Openly LGBTQ+ people in family or not.** Participants answered a question developed by the research team about LGBTQ+ in their families: "When you were growing up, how many openly LGBTQ+ people were there in your family?" Response options were 0, 1 or 2, 3 or 4, or 5 or more. Given the response distribution, this variable was recoded 0 (No openly LGBTQ+ people in family) and 1 (Having openly LGBTQ+ people in family).

**Demographic.** Demographic variables for participants included the following: Age (in years); gender identity (0 = cisgender; 1 = transgender and gender diverse); sexual orientation identity (0 = gay/lesbian; 1 = bisexual and/or pansexual; 2 = queer; 3 other identities [e.g., asexual, questioning]); racial/ethnic identity (0 = White; 1 = Asian or Pacific Islander; 2 = Black; 3 = Latinx; 4 = Multiracial; 5 = Native American); immigrant status (0 = US-born and not from an immigrant family; 1 = immigrant family background); and geographic area grow up in (0 = small town/rural), 1 (suburban), and 2 (large city/urban).

## Data analysis

Analyses were conducted in Stata/BE 18.0. Descriptive statistics, including central tendency, variability, and frequency distributions, were run. Main assumptions of linear regression, including linearity, homoscedasticity, normality of the error term, and no multicollinearity were tested using visual tests, Breusch-Pagan test, and variance inflation factors, and no issues that would preclude regression were found.

Next, guided by Fig 1, a series of multiple linear regression models were run to examine how family environment factors growing up (e.g., homophobic messages received from family; conservatism level in family growing up; openly LGBTQ+ people in family or not) were associated with LGBTQ+ adult's depression, anxiety, alcohol use, and IH. Openly LGBTQ+ people in family was a dichotomous variable, and other independent variables (IVs) and dependent variables (DVs) were continuous variables. After multilinear regression tests, the IH variable was further assessed to determine if there was a mediating relationship on family environment factors and behavioral health outcomes of depression, anxiety, and alcohol use. The first step to test a mediation effect was to examine whether the mediator (i.e.,IH) was associated with the IVs (i.e., homophobic messages received from family, conservatism level in family growing up, and openly

LGBTQ+ people in family or not). The IVs showing significant associations with the mediator were then regressed on each DV (i.e., depression, anxiety, and alcohol use) to see if there were significant associations between the mediator and DVs and between IVs and DVs. Six regression models were generated using Sobel tests. All the models controlled for age, racial/ethnic identity, gender identity, sexual orientation, immigrant background, and geographic area. Age was a continuous variable. The other demographic variables were categorical and were dummy coded for analyses. The Sobel tests calculations were conducted using an interactive calculation tool developed by Preacher and Leonardelli [81].

## Results

### Demographic characteristics

The sample included 499 LGBTQ+ adults, and the demographic characteristics are shown in Table 1. Ages ranged from 18 to 70 years, with an average age of 33.7 ($SD = 10.0$). Most participants identified as cisgender (86.0%) while 14.0% of them identified as transgender or gender diverse. Regarding sexual orientation, 60.9% identified as bisexual and/or pansexual, 29.9% identified as gay or lesbian, 8.4% identified as queer, and 4 (0.8%) participants had other sexual identities including demisexual or questioning. In terms of racial/ethnic identity, over 60% were White (64.8%), with additional participants identifying as Asian (6.2%), Black (14.6%), Latinx (11.8%), multiracial (0.4%), and Native American (2.2%). Most participants (85.4%) were U.S.-born and not from an immigrant family, and the sample was diverse on geographic background with 47.3% growing up in suburban area, 29.4% in small town or rural area, and 23.3% in large city or urban area.

### Demographics predicting behavioral health outcomes and IH

Compared to cisgender individuals, transgender people reported higher levels of depression ($\beta = .32$, $SE = 0.11$. $p < .01$). Bisexual/pansexual individuals had significantly higher levels of depression ($\beta = .31$, $SE = 0.08$, $p < .001$) and anxiety ($\beta = .14$, $SE = .06$, $p < .05$), and queer sexual orientation identity individuals had significantly higher depression ($\beta = .30$, $SE = .15$, $p < .05$) compared to gay/lesbian peers. Lastly, Asians or Pacific Islanders ($\beta = .59$, $SE = .19$, $p < .01$) and Black Americans ($\beta = .67$, $SE = .13$, $p < .001$) reported higher levels of IH compared to White peers. Latinx people reported significantly lower level of depression ($\beta = -.34$, $SE = .13$, $p < .01$) compared to White people. Native Americans reported significantly higher level of depression ($\beta = .52$, $SE = .25$, $p < .05$) and anxiety ($\beta = .51$, $SE = .18$, $p < .01$) compared to White peers. Results were shown in Table 2.

### Family environment factors predicting behavioral health outcomes and IH

Family environment factors in this study include homophobic messages received from family, conservatism level in family, and whether having openly LGBTQ+ people in family or not. Results showed that homophobic messages received from family significantly predict depression and anxiety, and conservatism level in family significantly predicted alcohol abuse and IH. As homophobic messages from family increased, levels of depression ($\beta = .21$, $SE = .05$, $p < .001$) and anxiety ($\beta = .14$, $SE = .04$, $p < .001$) increased significantly, but alcohol use ($\beta = .06$, $SE = .05$, $p = .280$) and IH ($\beta = .05$, $SE = .06$, $p = .347$) just increased slightly. As the conservatism level in family increased, levels of alcohol abuse ($\beta = .04$, $SE = .02$, $p < .05$) and IH ($\beta = .04$, $SE = .02$, $p < .05$) increased significantly, but depression ($\beta = .01$, $SE = .02$, $p = .575$) and anxiety level ($\beta = .01$, $SE = .01$, $p = .257$) did not. No significant difference in the DVs were found between people with and without openly LGBTQ+ people in the family growing up. Results were shown in Table 2.

### Mediation of IH

Among the three family environment factors, openly LGBTQ+ people in one's family did not have significant associations with any dependent variables; thus, it was not further examined for the indirect effect analysis. As results shown in Table 3, IH partially mediated the relationship between homophobic messages from family and depression ($z = 2.00$,

**Table 1. Participant demographics (n = 499).**

| Variable | Mean (*SD*) or *n* (%) |
| --- | --- |
| Internalized homonegativity (range = 1–7) | 2.1 (0.9) |
| Depression (range = 1–5) | 2.2 (0.9) |
| Anxiety (range = 1–4) | 1.8 (0.6) |
| Alcohol abuse (range = 1–5) | 1.5 (0.8) |
| Homophobic messages from family (range = 1–5) | 2.4 (0.8) |
| Conservatism level in family growing up (range = 1–9) | 1.2 (0.8) |
| Having openly LGBTQ+ people in family growing up | |
| Yes | 147 (29.5) |
| No | 352 (70.5) |
| Age | 33.7 (10.0) |
| Gender identity | |
| Cisgender | 429 (86.0) |
| Transgender and gender diverse | 70 (14.0) |
| Sexual orientation | |
| Gay or Lesbian | 149 (29.9) |
| Bisexual, Pansexual, or Bi/Pan+ | 304 (60.9) |
| Queer | 42 (8.4) |
| Other | 4 (0.8) |
| Racial/ethnic identity | |
| White | 323 (64.7) |
| Asian or Pacific Islander | 31 (6.2) |
| Black or African American | 73 (14.6) |
| Latinx | 59 (11.8) |
| Multiracial | 2 (0.4) |
| Native American | 11 (2.2) |
| Immigrant status | |
| U.S.-born and not from an immigrant family | 426 (85.4) |
| Immigrant family background | 73 (14.6) |
| Geographic area grew up in | |
| Small town or rural area | 147 (29.4) |
| Suburban area | 236 (47.3) |
| Large city or urban area | 116 (23.3) |

Notes: SD = Standard Deviation.

$p < .05$). IH also fully mediated the relationships between family conservatism and depression ($z = 3.35$, $p < .001$), family conservatism and anxiety ($z = 2.38$, $p < .05$), and family conservatism and alcohol abuse ($z = 2.65$, $p < .01$).

## Discussion

### Family environments factors and behavioral health outcomes

The results showed homophobic messages at family growing up predicts LGBTQ+ people's depression and anxiety level in adulthood, which aligns with literature that LGBTQ+ individuals who live in less accepting/affirming environments are at a heightened risk of experiencing negative behavioral health conditions, including depression, PTSD, and suicidal ideation [82,83]. Wang and others found that young adults exposed to homophobic messaging, both verbally and online,

 

**Table 2. Results of multivariate linear regression models predicting behavioral health outcomes (n = 499).**

| | Model 1: Depression β (SE) | Model 2: Anxiety β (SE) | Model 3: Alcohol abuse β (SE) | Model 4: IH β (SE) |
|---|---|---|---|---|
| Homophobic messages received from family | **0.21 (0.05)*** | **0.14 (0.04)*** | 0.06 (0.05) | 0.05 (0.06) |
| Conservatism level in family growing up | 0.01 (0.02) | 0.01 (0.01) | **0.04 (0.02)*** | **0.07 (0.02)*** |
| Openly LGBTQ+ people in family (Ref: No) | | | | |
| Yes | 0.06 (0.08) | 0.10 (0.06) | 0.13 (0.08) | 0.09 (0.09) |
| Age | **−0.01 (0.004)** | **−0.01 (0.003)*** | −0.001 (0.004) | 0.01 (0.004) |
| Gender identity (Ref: Cisgender) | | | | |
| Transgender | **0.32 (0.11)** | 0.11 (0.08) | −0.06 (0.11) | −0.14 (0.12) |
| Sexual orientation (Ref: Gay or Lesbian) | | | | |
| Bisexual, Pansexual, or Bi/Pan+ | **0.31 (0.08)*** | **0.14 (0.06)*** | 0.11 (0.09) | −0.09 (0.09) |
| Queer | **0.30 (0.15)*** | 0.21 (0.11) | 0.12 (0.15) | −0.19 (0.16) |
| Other | 0.66 (0.41) | 0.33 (0.30) | 0.11 (0.43) | 0.36 (0.47) |
| Racial/ethnic identity (Ref: White) | | | | |
| Asian or Pacific Islander | −0.16 (0.17) | −0.12 (0.13) | −0.28 (0.18) | **0.59 (0.19)** |
| Black or African American | −0.19 (0.11) | −0.06 (0.08) | 0.17 (0.12) | **0.67 (0.13)*** |
| Latinx | **−0.34 (0.13)** | −0.06 (0.09) | −0.001 (0.13) | 0.24 (0.14) |
| Multiracial | 0.63 (0.58) | 0.17 (0.43) | −0.64 (0.60) | −0.06 (0.64) |
| Native American | **0.52 (0.25)*** | **0.51 (0.18)** | 0.23 (0.26) | −0.23 (0.28) |
| Immigrant status (Ref: U.S.-born and not from an immigrant family) | | | | |
| Immigrant family background | 0.17 (0.12) | 0.06 (0.09) | −0.02 (0.12) | −0.16 (0.13) |
| Geographic area grew up in (Ref: small town or rural area) | | | | |
| Suburban area | −0.10 (0.09) | −0.09 (0.06) | −0.08 (0.09) | −0.06 (0.09) |
| Large city or urban area | −0.06 (0.11) | −0.10 (0.08) | 0.07 (0.11) | 0.06 (0.12) |
| Model fit | | | | |
| F | **5.75*** | **4.91*** | **1.75*** | **4.25*** |
| Adjusted $R^2$ | 0.132 | 0.112 | 0.024 | 0.094 |

* $p < .05$, ** $p < .01$, *** $p < .001$.

developed higher rates of severe depression, anxiety, and physical pain in emerging adulthood [84]. It should be noted that homophobic comments do not have to be directly communicated to LGBTQ+ youth for them to be harmful—just exposure to homophobic messages in the family environment can have subsequent negative effects.

The results showed that family conservatism significantly predicted alcohol abuse and IH. These results align with findings from Pacilli and Colleagues' study [85], which found a significant association between right-wing conservative political orientation and IH in a LGBTQ+ sample. Also, scholars have found that parents' political conservatism contributed to negative parental reactions to their LGBTQ+ children's coming out [86]. No existing literature was found on the influence of family conservatism and LGBTQ+ alcohol abuse, which warrants further research.

Extant research suggests that growing up with LGBTQ+ parents or relatives can offer benefits, such as a smoother coming-out processes, reduced concern about family rejection, reduced isolation, and better connection to LGBTQ+ communities and cultures [64,65], which are associated with better behavioral health outcomes [15,87] and lower IH [74].

**Table 3. Mediation effect results of IH using Aroian Sobel Tests (n = 499).**

| Testing steps | | Model 1 | Model 2 | Model 3 | Model 4 | Model 5 | Model 6 |
|---|---|---|---|---|---|---|---|
| | IV | Homophobic messages | Homophobic messages | Homophobic messages | Conservatism level | Conservatism level | Conservatism level |
| | DV | Depression | Anxiety | Alcohol abuse | Depression | Anxiety | Alcohol abuse |
| 1 | Beta$_A$ (*SE*) between IV and Mediator | 0.12 (0.05)* | 0.12 (0.05)* | 0.12 (0.05)* | 0.07 (0.02)*** | 0.07 (0.02)*** | 0.07 (0.02)*** |
| 2 | Beta$_B$ (*SE*) between Mediator and DV | 0.20 (0.04)*** | 0.08 (0.03)** | 0.15 (0.04)*** | 0.21 (0.04)*** | 0.09 (0.03)** | 0.14 (0.04)** |
| | Beta$_C$ (*SE*) between IV and DV | 0.20 (0.05)*** | 0.15 (0.04)*** | 0.08 (0.05) | 0.01 (0.02) | 0.02 (0.01) | 0.03 (0.02) |
| 3 | Aroian Sobel test statistic Beta (*SE*) | 2.00 (0.01)* | 1.72 (0.01) | 1.86 (0.01) | 3.35 (0.01)*** | 2.38 (0.003)* | 2.65 (0.004)** |
| | Mediation effect | Partial mediation | None | None | Fully mediation | Fully mediation | Fully mediation |

* *p* < .05, ** *p* < .01, *** *p* < .001.

However, our study found no association between having openly LGBTQ+ family members and behavioral health outcomes or IH. One possible explanation is that the presence of openly LGBTQ+ individuals in the family is not an essential component of a supportive family environment. Supportive and affirming family dynamics can manifest through acceptance, open communication, inclusive language, and consistent and explicit support [88,89], regardless of whether there are open LGBTQ+ family members. These family elements have been shown to play critical roles in shaping LGBTQ+ youth's behavioral health and internalized attitudes [15,87]. For example, Stone and Colleagues identified supportive behaviors for LGBTQ+ youth [65], including LGBTQ+ family members helping youth navigate family communication norms, introducing LGBTQ+ cultures, and helping mediate conflict between LGBTQ+ youth and parents. Future research should more deeply examine how openly LGBTQ+ family members interact in various ways in family contexts that can affect LGBTQ+ youth.

## IH as a mediator

While initial regression analysis found that family conservatism was associated with alcohol abuse, mediation results suggest that IH fully mediated the relationship between family conservatism and all three behavioral health outcomes. In other words, family conservatism may have adverse effects on LGBTQ+ individuals' depression, anxiety, and alcohol abuse only when channeled through IH. No prior studies were located that examined IH as a mediator between family conservatism growing up and the outcomes of depression, anxiety, and alcohol abuse. Nonetheless, politics has played a significant role in the lives of LGBTQ+ people for decades, with LGBTQ+ rights and discrimination being the focus of political movements and the enactment of legislation via political parties [90,91]. Multiple studies (with the general population and LGBTQ+ participants) have found that political conservatism is associated with negative attitudes about LGBTQ+ communities [50–54]. Therefore, growing up in a family with politically conservative ideologies likely contributes to IH, and there is substantial research connecting IH and behavioral health problems [71–73,92,93]. Given that political conservatism in the family context is a complex construct that could encompass multiple elements, more research is needed in this area.

Our findings that IH partially mediates the relationship between family homophobic messaging and depression fill a gap in the literature as there is a lack of evidence about the mediation role of IH between family homophobic messages and behavioral health outcomes for LGBTQ+ youth. Our findings do align studies showing IH as a mediator between lesbians and gay men's experiences of general experiences with discrimination and symptoms of depression [94]. Moreover, IH has been positively associated with family sexual stigma and mental health problems among LGBTQ+ people [95,96]. These results from our findings along with previous literature raise important concerns on the influence of family

environment given that IH can develop over long periods of time despite avoiding interactions with homophobic family members during adulthood [96].

## Implications for practice

This study underscores the critical role of childhood family environments in shaping the behavioral health and internalized self-concepts of LGBTQ+ adults. Interventions aimed at reducing behavioral health disparities among LGBTQ+ populations should focus on addressing the adverse effects of unsupportive family environments, such as homophobic language and conservative political views. Incorporating family dynamics and inclusive practices into therapy, support programs, and educational initiatives would help mitigate the negative impacts of childhood environmental homophobia. While existing efforts, such as school-based Gender-Sexuality Alliances and the Family Acceptance Project, and have demonstrated positive impacts on the well-being of LGBTQ+ individuals [97,98], these programs primarily serve those who are already aware of their own or their children's LGBTQ+ identities. Expanding such initiatives to include the general population could foster greater awareness and inclusivity, thereby reducing the prevalence of environmental homophobia more broadly. Moreover, programs supporting families with LGBTQ+ members, such as support groups offered by Parents, Families, and Friends of Lesbians and Gays (PFLAG), could be further developed more to encourage more families to engage in creating inclusive environments for LGBTQ+ children. In addition to developing inclusive programs, when working with LGBTQ+ individuals, clinicians should explore with them how their beliefs and perception about LGBTQ+ identities. One's thoughts and feelings about being an LGBTQ+ individual in hetero- and cis-normative societies impacts their stress and health outcomes [4,99,100]. Interventions on reducing IH should be emphasized in research and practice to provide supportive service for LGBTQ+ populations.

Data for this study were collected during the COVID-19 pandemic, a period that may strengthen stigmatization existing pre-pandemic and worsen LGBTQ+ individual's behavioral health due to undesired family interactions, social isolation, or political polarization [101]. These unique circumstances demonstrated the needs for accessible and crisis-responsive support or service systems that can address LGBTQ+-related stressors, especially within the family context.

## Limitations

The present study has some limitations, which should be addressed in future research. First, regarding generalizability, the participants were primarily cisgender individuals (86.0%), which may not fully reflect the range of life experiences of gender-diverse people. Some items in the homophobic message scale were modified to include gender-diverse populations, but the survey still lacks questions specifically related to gender diversity or transphobic situations. In addition, data for this study were collected during the COVID-19 pandemic. This unique circumstance may have influenced participants' experiences of family interactions, minority stress, and mental health. For example, LGBTQ+ people may have lost access to LGBTQ+ communities and moved back into non-affirming homes, increasing the likelihood of tense family situations. As such, the findings may not fully generalize to non-pandemic conditions.

Second, the participants were adults with ages ranging from 18 to 70 years but were asked to recall the family environment growing up, which may involve potential recall bias. Furthermore, the researchers who designed the survey did not provide a fixed definition of "family" in the survey. Therefore, the question relies on participants' understanding of family. While the use of "when you were growing up" in the item implies that the question refers to the family of origin, it still could be interpreted differently by participants (e.g., immediate family vs. extended family).

Third, regarding the analysis, our data were cross-sectional, so no causal inferences can be made about whether family environments and IH lead to behavioral health outcomes. Moreover, we conducted multiple regression models without applying formal corrections for multiple testing. Although the models were guided by theoretical framework and prior empirical findings to reduce the concern of inflated Type I error, we acknowledge this as a limitation in the data analysis. Last, due to data availability, information of gender identity disaggregated into cisgender-men and cisgender-women, as well as more specific breakdown (e.g., bi/pansexual), was not included in the analysis.

## Future research

Future research should use longitudinal study designs to better understand how childhood family environments influence LGBTQ+ individuals' behavioral health and IH over time and help establish causal relationships. Studies should include more gender-diverse participants and more detailed demographic breakdowns, such as sexual orientation subgroups. Exploring the interplay of diverse family structures, such as stepfamilies or same-sex parent families, and intersecting identities, such as racial/ethnic identity, gender identity, and socioeconomic background, could provide valuable insights into these complex dynamics. Additionally, developing and applying clear definitions of family will be important to ensure that findings are interpretable within specific contexts. Further investigations are also needed to examine the potential intersectional influences of age, race/ethnicity, gender identity, sexual orientation, and immigrant background on the mediating role of internalized homonegativity and behavioral health outcomes. These social identities may interact to shape both the experience of stigma and the coping mechanisms to LGBTQ+ individuals. Furthermore, based on the results of this study, more research is needed to explore how openly LGBTQ+ family members affect LGBTQ+ youth and what aspects of political views within the family context impact the behavioral health of LGBTQ+ individuals. Lastly, investigating the protective effects of affirming family environments may inform strategies to promote equitable behavioral health outcomes for LGBTQ+ communities. Finally, researchers should consider implementing corrections for multiple testing when conducting numerous statistical models, particularly in exploratory analyses, to enhance the robustness of findings.

## Supporting information

**S1 File. Dataset for PLOS One.**
(XLSX)

## Author contributions

**Conceptualization:** Pin-Chen Chiang, Ankur Srivastava.

**Data curation:** William J. Hall.

**Formal analysis:** Pin-Chen Chiang.

**Funding acquisition:** William J. Hall.

**Investigation:** William J. Hall.

**Project administration:** Pin-Chen Chiang, William J. Hall.

**Resources:** William J. Hall.

**Supervision:** Ankur Srivastava, William J. Hall.

**Validation:** William J. Hall.

**Visualization:** Pin-Chen Chiang.

**Writing – original draft:** Pin-Chen Chiang, Yinuo Xu, Denise Yookong Williams, Ankur Srivastava, Jake A. Leite, Adam R. Englert, William J. Hall.

**Writing – review & editing:** Pin-Chen Chiang, Yinuo Xu, Denise Yookong Williams, Ankur Srivastava, William J. Hall.

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
