## [Decision Letter · Decision Letter 0]

PONE-D-25-05634The Relationship between family environments growing up and behavioral health among LGBTQ+ adults: The mediating role of internalized homonegativityPLOS ONE

Dear Dr. Chiang,

Thank you for submitting your manuscript to PLOS ONE. After careful consideration, we feel that it has merit but does not fully meet PLOS ONE’s publication criteria as it currently stands. Therefore, we invite you to submit a revised version of the manuscript that addresses the points raised during the review process.<article class="text-token-text-primary w-full" data-scroll-anchor="false" data-testid="conversation-turn-2" dir="auto">

**The reviewers agree that the paper makes a valuable contribution. To strengthen the manuscript further, it would be appropriate to further incorporate relevant minority stress and microaggression literature and addressing analytical gaps, particularly concerning gender identity, sexual orientation subgroups, and the role of age in the context of your study. Furthermore, please review comments form Reviewer 1 around clarifying methodological aspects, particularly around scale scoring and variable definitions. It is also important to consider expanding the introduction and discussion to include global disparities, school-based supports, and the context of COVID-19.**

We look forward to receiving your revised manuscript.

Kind regards,

Daniel Demant, PhD, MPH, GradCertHEd, BAppSocSc

Academic Editor

PLOS ONE

**Journal Requirements:**

1. When submitting your revision, we need you to address these additional requirements. Please ensure that your manuscript meets PLOS ONE's style requirements, including those for file naming. The PLOS ONE style templates can be found at https://journals.plos.org/plosone/s/file?id=wjVg/PLOSOne_formatting_sample_main_body.pdf and https://journals.plos.org/plosone/s/file?id=ba62/PLOSOne_formatting_sample_title_authors_affiliations.pdf 2. Please provide additional details regarding participant consent. In the ethics statement in the Methods and online submission information, please ensure that you have specified (a) whether consent was informed and (b) what type you obtained (for instance, written or verbal, and if verbal, how it was documented and witnessed). If your study included minors, state whether you obtained consent from parents or guardians. If the need for consent was waived by the ethics committee, please include this information. If you are reporting a retrospective study of medical records or archived samples, please ensure that you have discussed whether all data were fully anonymized before you accessed them and/or whether the IRB or ethics committee waived the requirement for informed consent. If patients provided informed written consent to have data from their medical records used in research, please include this information. 3. Thank you for stating in your Funding Statement: Research reported in this article was supported by the National Institute of Minority Health and Health Disparities of the National Institutes of Health (https://www.nimhd.nih.gov/) under award number R21MD012687. W.H. is the author who received this award. The funders had no role in study design, data collection and analysis, decision to publish, or preparation of the manuscript. Please provide an amended statement that declares *all* the funding or sources of support (whether external or internal to your organization) received during this study, as detailed online in our guide for authors at http://journals.plos.org/plosone/s/submit-now.  Please also include the statement “There was no additional external funding received for this study.” in your updated Funding Statement. Please include your amended Funding Statement within your cover letter. We will change the online submission form on your behalf. 4. Thank you for stating the following in the Acknowledgments Section of your manuscript: Research reported in this article was supported by the National Institute of Minority Health and Health Disparities of the National Institutes of Health under award number R21MD012687. We note that you have provided funding information that is not currently declared in your Funding Statement. However, funding information should not appear in the Acknowledgments section or other areas of your manuscript. We will only publish funding information present in the Funding Statement section of the online submission form. Please remove any funding-related text from the manuscript and let us know how you would like to update your Funding Statement. Currently, your Funding Statement reads as follows: Research reported in this article was supported by the National Institute of Minority Health and Health Disparities of the National Institutes of Health (https://www.nimhd.nih.gov/) under award number R21MD012687. W.H. is the author who received this award. The funders had no role in study design, data collection and analysis, decision to publish, or preparation of the manuscript.  Please include your amended statements within your cover letter; we will change the online submission form on your behalf. 5. In the online submission form, you indicated that your data is available only on request from a third party. Please note that your Data Availability Statement is currently missing the name of the third party contact or institution and contact details for the third party, such as an email address or a link to where data requests can be made. Please update your statement with the missing information. 

**Additional Editor Comments:**

An important aspect that I found that was not addressed is the potential intersectional influence of race, ethnicity,and immigrant background on the mediating role of internalised homonegativity and behavioural health outcomes. Your sample presents significant demographic diversity and includes these variables in regression models, but further interpretation of how these intersecting identities may compound or mitigate the impact of familial homophobia and internalised stigma could provide valuable insights. If this is beyond the scope of your research, this should be discussed clearly as to why.

Reviewers' comments:

Reviewer's Responses to Questions

**Comments to the Author**

1. Is the manuscript technically sound, and do the data support the conclusions?

Reviewer #1: Yes

Reviewer #2: Yes

2. Has the statistical analysis been performed appropriately and rigorously? 

Reviewer #1: Yes

Reviewer #2: Yes

3. Have the authors made all data underlying the findings in their manuscript fully available?

Reviewer #1: Yes

Reviewer #2: No

4. Is the manuscript presented in an intelligible fashion and written in standard English?

Reviewer #1: Yes

Reviewer #2: Yes

5. Review Comments to the Author

**Reviewer #1:**  This well-structured and insightful study explores the relationship between family environments growing up and behavioral LGBTQ+ adults with the mediation of internalized homonegativity. This research appears highly relevant and opens avenues for future investigations in other settings.

I support the publication of this manuscript with minor revisions:

1. Manuscript:

• Please ensure that page numbers and/or line numbers are correctly indicated.

2. Introduction:

• Be more precise when you used “world” in the sentence “Even though societies adopt more policies protecting LGBTQ+ rights and discussing LGBTQ+ issues openly more than before, heteronormativity and cisnormativity remain dominant in the world and affect social systems and LGBTQ+ people’s health and development (Mills-Koonce et al., 2018; Van Bergen et al., 2021). Maybe you talk about countries. It could be interesting to add that some of them criminalize LGBTQ+ people. Please detail in which ways it impact heath and development of LGBTQ+ people.

• Add reference for : “IH is also considered a consequence of heteronormative societal beliefs and affects one’s behavioral Health »

• In the Family Environments for LGBTQ+ section

o Maybe you could add a subsection title as “Social environment” for first paragraph as a broader picture or modify the section by “Social and Family Environments for LGBTQ+”

o “Despite important findings about influences of family and caregiver interactions with LGBTQ+ children’s behavioral health outcomes, existing studies largely focus on individual-level factors (e.g., caregiver factors).” Please add examples of individual level factors and references

3. Methods:

• Please define precisely which scores are calculated for each scale. For example, you add final scores ranged from 5 to 25 for Homophobic messages received from family. Add for others scales and add these informations in table 1 for each scale result to understand your results.

• Openly LGBTQ+ people in family or not :

o How do you define family to answer this question : When you were growing up, how many openly LGBTQ+ people were there in your family?”

o Response options were 0, 1 or 2, 3 or 4, or 5 or more. Given the response distribution, this variable was recoded 0 (0) and 1 (1 or more). Give example for this variable recoding. By the way, it is not defined in results section.

• Data analysis : did you think of a correction for multiple models applied to data ?

4. Results:

• Range for scales in table 1 (see method section)

• Openly LGBTQ+ people in family or not : not define (see method section)

• Pages 13-14 : please put table in one page

5. Discussion:

• You discussed results about having openly LGBTQ+ family member in page 17 but don’t mentioned it in limitations. Please discussed this definition of family and variable recoding

• Please discussed study context of covid and potential implications of this specific period in your study. For example responses of participants if they coming back to their family and locked with them in that period.

**Reviewer #2:**  This paper offers a useful contribution to the research literature on the antecedents of mental health issues experienced by LGBTQ+ people in familial homophobia, and in particular, how internalized homophobia mediates this relationship. There remain a few questions that should be addressed to shape up the paper into final form:

While the review of the research literature covers many relevant studies, there needs to be some reference to the “minority stress” and homophobic “micro-aggression” research literatures which consolidate many of these findings around a few central concepts.

Some analytical questions:

• While the study reports demographic rates for cis-gender/trans and sexual identities, there is curiously no mention of adult gender identity distinguishing men and women. Was gender used in the analysis? Were there any gender differences in either the familial or mental health variables? Related to this, gay and lesbian cannot simply be treated as a single unitary identity. Were there any differences between gay and lesbian categories? Also, what is the gender breakdown of bi/pansexual respondents? Given that the majority of the sample identified as bi/pansexual and had significantly higher rates of anxiety and depression, did gay, lesbian, or trans people present different profiles regarding IH or indicators of familial homophobia?

• Given that increasing age was associated with lower anxiety/depression (in accord with other studies) and age can be associated with sexual identity, particularly bisexual identity, was age controlled for in reporting rates of anxiety or depression? In other words, how much of these bisexual rates are accounted for by age?

In the Implications section, which gestures in the direction of “inclusive practices in therapy, support programs, and educational initiatives [that] would help mitigate the negative impacts of childhood environmental homophobia,” it should be noted that there is significant research on the supportive effects of Gay/Straight Alliances (or equivalent) and anti-bullying programs in schools in improving the lives of LGBTQ+ youth.

Editorial notes:

• intro: should be “Trevor Project”

• family environment section: should be “health outcomes in adulthood have been under-researched”

• p 15 should be “LGBTQ+ people in one’s family did”

6. PLOS authors have the option to publish the peer review history of their article (what does this mean? ). If published, this will include your full peer review and any attached files.

**Do you want your identity to be public for this peer review?** For information about this choice, including consent withdrawal, please see our Privacy Policy .

Reviewer #1: No

Reviewer #2: **Yes: ** Barry D Adam

---

## [Author Response · Author response to Decision Letter 1]

9 Jun 2025

Thank you for giving us the opportunity to revise and resubmit this manuscript. Please see the revision table with comments for responses.

---

## [Editor Report · Decision Letter 1]

The Relationship between family environments growing up and behavioral health among LGBTQ+ adults: The mediating role of internalized homonegativity

PONE-D-25-05634R1

Dear Dr. Chiang,

We’re pleased to inform you that your manuscript has been judged scientifically suitable for publication and will be formally accepted for publication once it meets all outstanding technical requirements.

Kind regards,

Daniel Demant, PhD

Academic Editor

PLOS ONE

Additional Editor Comments (optional):

All feedback from the reviewers has been addressed sufficiently.
---

## [Editor Report · Acceptance letter]

PONE-D-25-05634R1

PLOS ONE

Dear Dr. Chiang,

I'm pleased to inform you that your manuscript has been deemed suitable for publication in PLOS ONE. Congratulations! Your manuscript is now being handed over to our production team.

Kind regards,

on behalf of

Associate Professor Daniel Demant

Academic Editor

PLOS ONE